# Mild Muscle Mitochondrial Fusion Distress Extends *Drosophila* Lifespan through an Early and Systemic Metabolome Reorganization

**DOI:** 10.3390/ijms222212133

**Published:** 2021-11-09

**Authors:** Andrea Tapia, Martina Palomino-Schätzlein, Marta Roca, Agustín Lahoz, Antonio Pineda-Lucena, Víctor López del Amo, Máximo Ibo Galindo

**Affiliations:** 1Centro de Investigación Príncipe Felipe, 46012 Valencia, Spain; atapia@cipf.es (A.T.); mpalomino@cipf.es (M.P.-S.); 2Analytical Unit, Medical Research Institute-Hospital La Fe, Av. Fernando Abril Martorell 106, 46026 Valencia, Spain; marta_roca@iislafe.es (M.R.); agustin.lahoz@uv.es (A.L.); 3Biomarkers and Precision Medicine Unit, Medical Research Institute-Hospital La Fe, Av. Fernando Abril Martorell 106, 46026 Valencia, Spain; 4Molecular Therapeutics Program, Centro de Investigación Médica Aplicada, Universidad de Navarra, 31008 Pamplona Spain; apinedal@unav.es; 5Section of Cell and Developmental Biology, University of California, San Diego, CA 92093, USA; 6Instituto Interuniversitario de Investigación de Reconocimiento Molecular y Desarrollo Tecnológico (IDM), Universitat Politècnica de València, 46022 Valencia, Spain; 7UPV-CIPF Joint Unit Disease Mechanisms and Nanomedicine, 46012 Valencia, Spain

**Keywords:** mitohormesis, metabolomics, lifespan, *Drosophila*, insulin pathway

## Abstract

In a global aging population, it is important to understand the factors affecting systemic aging and lifespan. Mitohormesis, an adaptive response caused by different insults affecting the mitochondrial network, triggers a response from the nuclear genome inducing several pathways that promote longevity and metabolic health. Understanding the role of mitochondrial function during the aging process could help biomarker identification and the development of novel strategies for healthy aging. Herein, we interfered the muscle expression of the *Drosophila* genes *Marf* and *Opa1*, two genes that encode for proteins promoting mitochondrial fusion, orthologues of human *MFN2* and *OPA1*. Silencing of *Marf* and *Opa1* in muscle increases lifespan, improves locomotor capacities in the long term, and maintains muscular integrity. A metabolomic analysis revealed that muscle down-regulation of *Marf* and *Opa1* promotes a non-autonomous systemic metabolome reorganization, mainly affecting metabolites involved in the energetic homeostasis: carbohydrates, lipids and aminoacids. Interestingly, the differences are consistently more evident in younger flies, implying that there may exist an anticipative adaptation mediating the protective changes at the older age. We demonstrate that mild mitochondrial muscle disturbance plays an important role in *Drosophila* fitness and reveals metabolic connections between tissues. This study opens new avenues to explore the link of mitochondrial dynamics and inter-organ communication, as well as their relationship with muscle-related pathologies, or in which muscle aging is a risk factor for their appearance. Our results suggest that early intervention in muscle may prevent sarcopenia and promote healthy aging.

## 1. Introduction

Aging is a natural process governed by the loss of organism homeostasis due to different time-dependent molecular imbalances. As the aging population increases worldwide, there is an emerging interest in knowing the mechanisms that lead to longevity, not only to increase it, but also to promote a healthy aging.

Mitochondria are one of the proposed aging hallmarks [1] acting as an environmental sensor and unfettering a complex response from the nuclear genome under stress conditions. However, mitochondrial shape modulation and its effect on lifespan remains partially unexplored. The mitochondrial network undergoes morphological changes by fission and fusion that affect many processes, such as mitochondrial distribution, mitophagy, and mitochondrial biogenesis. This process, known as mitochondrial dynamics, has been related with aging [2], metabolism [3] and apoptosis [4] to compensate for perturbed organism homeostasis. In order to adapt to the mentioned physiological derangements, mitochondrial shape is highly dynamic, mainly due to fusion and fission processes that are controlled by a specific set of proteins [5]. Mitochondrial fusion is regulated by two proteins: Mitofusin 1/2 tethers the mitochondria from their outer membranes, and *Opa1* finishes the process by merging the organelle inner membrane. In turn, *Drp1* governs mitochondrial fission creating a contractile ring that divides the mitochondria [6]. In addition, Dynamin-2 has also been reported as an indispensable protein for mitochondrial fragmentation machinery [7]. 

Aging is a progressive process towards loss of cell function, and susceptibility to degenerative diseases; interestingly, manipulation of signaling pathways in a tissue-specific manner can prolong or shorten life span [8,9]. Muscle tissue is characterized by a high abundance of mitochondria due to a high energy demand, playing a central role in metabolism and contributing to fine-tuned control of systemic metabolite levels for the organism′s homeostasis [3]. Indeed, muscle is a key organ in regulating systemic aging [10]. In particular, *Drosophila* muscle perturbation produces systemic effects on aging: down-regulation of the electron transport chain (ETC) extends lifespan by a systemic insulin signaling blockage [11], and FOXO/4EBP signaling in the same tissue also regulates protein turnover during aging [12]. More recently, *Drp1* muscle overexpression was shown to increase lifespan in *Drosophila* [13]. These observations demonstrate that muscle manipulation can lead to lifespan increase, and therefore delay the natural aging process. Although several works shed light on the mitochondrial dynamics influence in the aging process, we lack studies which perform tissue-local interventions while exploring their impact on systemic aging.

For this reason, we wondered if impaired mitochondrial fusion only in the muscle could influence systemic aging in *Drosophila*. We hypothesize that an alteration of mitochondrial fusion in muscle tissue could cause autonomous effects in this tissue while modulating metabolome homeostasis in distant tissues to regulate lifespan.

To test this hypothesis, we first down-regulated the *Marf* and *Opa1* canonical mitochondrial fusion-regulating genes in the *Drosophila* muscle. Then, we analyzed separately the metabolome of *Drosophila* head, thorax and abdomen tissues by using nuclear magnetic resonance (NMR). NMR is a simple and reproducible method for metabolomic profiling [14]. This approach has revealed age-associated changes in skeletal muscle metabolites [15], and relevant differences in several metabolites in human plasma associated with muscle quality [16].

Metabolomics studies were performed in these tissues for both mutant and control flies, and combined with behavioral assays, histological analysis and quantitative PCR (qPCR). Our results show that altered levels of *Marf* and *Opa1* only in muscle extended *Drosophila* life span, improved its locomotor capacities in the long-term and delayed muscle wasting. Most importantly, a correlation could be established between the reduction in mitochondrial fusion and an altered metabolic profile in muscle, coupled with a non-autonomous reorganization of the metabolome in the head and abdomen.

## 2. Results

### 2.1. Marf and Opa1 Down-Regulation in the Muscle Increase Lifespan and Preserves Locomotor Capacities in the Long Term

In order to know if impaired mitochondrial fusion in the muscle could influence systemic aging, we down-regulated the *Marf* and *Opa1* genes in the *Drosophila* muscle by using the *Gal4*/*UAS* technique for directed expression of transgenes [17]. The *Mhc-Gal4* transgenic construct drives expression of the yeast *Gal4* transcription factor under the control of the *Myosin heavy chain* (*Mhc*) gene promoter in differentiated muscle [18]. We combined the *Mhc-Gal4* driver with either a *UAS-Marf^RNAi^* or a *UAS-Opa1^RNAi^* transgenic line to down-regulate (knock-down; KD) either gene exclusively in the adult muscle. The specificity of both constructs has been validated previously, ruling out off-target effects [19]. Henceforth, we will call these two experimental genotypes *Marf^KD^* and *Opa1^KD^*, respectively. In every subsequent experiment, the control genotype harbored the *Mhc-Gal4* driver, but no *UAS* construct.

Our objective was to differentiate between autonomous and non-autonomous effects of the RNAi genotypes, but there is considerable technical difficulty in dissecting enough adult material to perform a metabolomic analysis. Therefore, our experimental approach was to dissect the body in three anatomical parts: the head, containing mostly neuronal cells from the central and peripheral nervous systems; the thorax, mostly composed by muscle tissue; and the abdomen occupied by the gonads, the digestive system and associated organs, and the fat body. It is true that these three samples are not histologically homogenous (i.e., the thorax also has some nervous and digestive system, and the head and abdomen have some muscle tissue); but the subsequent metabolomic analyses clearly showed that they are metabolomically different, and that muscle-specific metabolites such as β-alanine are mostly present in thorax extracts. This evidence confirmed that our experimental design was a good approach towards comparing autonomous versus non-autonomous effects (see below).

To ensure that the expression of the RNAi constructs resulted in reduced levels of the target transcripts, we first dissected fly thoraces from our experimental flies and performed qPCR to measure transcript levels. In both cases, *Marf* and *Opa1* mRNA levels upon RNAi led to a significant reduction in the expression of the transcripts (Figure 1A). Specifically, a ~60% reduction in transcript levels for both genotypes was detected. Interestingly, we also observed a non-specific and statistically less significant reduction in the other transcript (Figure 1A). This result validates the KD of *Marf* and *Opa1* genes and reveals a cross-regulation of these genes.

We first performed longevity assays in order to evaluate whether *Marf* and *Opa1* muscle down-regulation affects *Drosophila* lifespan. Indeed, the lifespan of *Marf^KD^* and *Opa1^KD^* was extended in comparison with control flies (Figure 1B, *p* value Mantel-Cox test < 0.0001). Regarding life expectancy (mean ± SD), it was 83 ± 2.08 days for the control, 89 ± 0.58 days for *Marf^KD^* and 92 ± 1.52 days for *Opa1^KD^*. The maximum lifespan for control flies was 98 days, however, both our experimental genotypes presented maximum lifespan of 109 (Figure 1B). This result indicates that muscle mitochondrial fusion perturbation by down-regulation of both *Marf* and *Opa1* genes influences *Drosophila* lifespan.

In order to identify possible effects on muscle function due to the mitochondrial alteration, we monitored negative geotaxis using the climbing assay at three time points, 10-, 30- and 60-day old flies. In addition to extended lifespan, flies of the *Marf^KD^* and *Opa1^KD^* genotypes showed a better preservation of the ability to climb the vial in the long term (Figure 1C), which suggests that the mitochondrial fusion distress could have a beneficial effect on locomotor capacities. To explore structural muscle integrity, we performed semi-thin sections of the indirect flight muscles in the thorax. Interestingly, we did not detect significant muscular degeneration in any of the genotypes at 65 days of age (Figure 1D). In order to study the fine morphology of the muscles, we performed transmission electron microscopy of ultra-thin longitudinal sections (Figure 1E). As expected, mitochondria of the *Marf^KD^* muscles had a marked reduction in size. This effect is less evident in the *Opa1^KD^* flies, but we could identify another phenotype of *Opa1* loss of function, the loss of structure of the mitochondrial cristae [20]. In contrast, the structure of the myofibrils is not affected.

In summary, the anti-aging effect induced by *Marf^KD^* and *Opa1^KD^*, manifesting in increased lifespan and preserved climbing capacity, suggests that *Drosophila* muscle could trigger a systemic response in the mutants conferring such a phenotype. To decipher the unexplored relationship between lifespan and systemic metabolic state under muscle mitochondrial fusion imbalance, we devised the following strategy. We analyzed the three anatomical regions as a proxy for autonomous (thorax) versus non-autonomous (head and abdomen) effects of the muscular alteration. We investigated the metabolomic profiles from each anatomical part at the age of 30 days, where all genotypes present a 100% survival rate, and at 65 days, when control flies started to die, but *Marf^KD^* and *Opa1^KD^* genotypes still had a 100% survival rate (Figure 1B). Then, the anatomical regions of the three genotypes at both ages were processed to obtain the corresponding polar and non-polar extracts, which were subsequently analysed by NMR. In all 36 NMR analyses, we performed six independent biological replicates with material from 20 individuals. After the quality control and analyses, one of these failed to produce a good enough spectrum, the polar extract nº 5 of young *Opa1^KD^* thoraces. Therefore, in this class we have five independent spectra instead of six.

### 2.2. The Different Anatomical Regions Have a Characteristic Metabolomic Profile

To obtain a general overview, we first explored how the metabolic profile differs between the three anatomical regions, reflecting their histological and biological properties. To this end, principal component analysis (PCA) of the samples’ metabolic profile from the three anatomical regions was performed, irrespective of their genotype and age. This kind of unsupervised multivariate analysis reduces the dimensionality of the large metabolic data table into principal components without considering classification labels of samples, reflecting the main variability trends between samples in an objective way. Our PCA model displayed metabolic differences between particular tissues as they clustered as independent groups (Figure 1F). The corresponding loading plot provided information about the metabolites that were characteristic for each anatomical region (Figure 1G). We observed an increased abundance of metabolites related to energy metabolism such as energetic nucleotides (NAD^+^, ATP and ADP) and TCA cycle intermediates (citrate, fumarate and succinate) in the head extracts. Additionally, high levels of neurotransmitters (glutamate and aspartate) and *N*-acetylaspartate (a neuronal osmolyte) were highly present in this anatomical region. The abdomen, which contains the digestive organs, the fat body, and the gonads, stood out for a high presence of molecules related to sugar and fat metabolism (glucose, trehalose and glycerol). Finally, in thorax, the region that contains the highest proportion of muscle tissue, we detected high amounts of amino acids, especially β-alanine, a muscle-specific metabolite [21], and its related species alanine and phenylalanine, as well as taurine.

In summary, the metabolic profile of the three different anatomical regions presented clear differences, reflecting their different function, histology and metabolism, and validates our assumption about the histological representativity of the different anatomical regions.

### 2.3. All Anatomical Regions Change Their Metabolomic Profile with Age

Next, we investigated how the metabolome of the three different anatomical regions changed with the age of the fly. To this end, we compared the metabolic profile of the head, thorax and abdomen in 30- and 65-day old flies, irrespectively of their genotype (control, *Marf^KD^* or *Opa1^KD^*). The resulting PCA loading plots showed a clear separation between 30- and 65-day-old samples in all three anatomical regions (Figure 2). It is worth mentioning that this distinction was much more evident in the thorax, suggesting that the muscle is a crucial tissue in systemic aging.

The corresponding PCA loading plots reflected which metabolic changes took place with age (Figure 2, with metabolite labels in Appendix A). In general, polar metabolites experienced a more significant alteration than lipids, especially in the thorax. As a general tendency, we observed that the proportion of sugar molecules increased, while organic acids were more prone to decrease. For instance, the tricarboxylic acid (TCA) cycle components decreased with age: we found lower levels of fumarate in all three regions of the 65-day-old animals. In the abdomen, citrate also decreased. Additional energy metabolism metabolites also seemed to be altered with age. We observed a marked decrease in ATP, ADP and NAD^+^ in all three regions, reinforcing an alteration in mitochondrial homeostasis, as previously described for human and rodent cells [22]. On the other hand, some amino acids increased while others decreased.

Regarding the non-polar phase, we noticed relevant changes in lipid levels: A general age-related increase in triacyclglycerols and phospholipids; and a head-specific trend to increased esterified lipids. Regarding the unsaturated lipids, the direction was towards a reduction in the thorax and more markedly in the head. In general, changes in non-polar metabolites were always more evident in head extracts.

### 2.4. Genotype-Specific Changes Are More Evident in Younger Flies

Since the *Marf^KD^* or *Opa1^KD^* genotypes influenced systemic aging, and therefore lifespan, we would expect specific metabolic differences in each one of the three body regions compared to control flies. To find out anatomically-specific metabolomic differences between the three genotypes, we performed PCA analysis by pooling the data by body region and age (Figure 3). This analysis revealed that *Marf^KD^* or *Opa1^KD^* 30-day-old flies were different compared with the control group (Figure 3A–C). However, we did not obtain a similar scenario at 65 days of age, probably due to greater intra-group variability (Figure 3D–F). To confirm these differences, we performed orthogonal partial least squares (OPLS) analysis by pairs, comparing each mutant genotype with the control. This kind of supervised discriminant analysis focuses only on differences between groups. Thus, we eliminate the general variability within sample groups that can mask significant discriminant changes. In this case, we could build robust models for the six pairwise comparisons at 30 days (Appendix A), but not at 65 days (Appendix A) reinforcing the idea of greater variability at this stage. Overall, the metabolic profiles of 30-day-old individuals were different by genotype and body region; in contrast, we could not identify significant differences between genotypes in older flies.

As differences in survival rates started at around 65 days (Figure 1B), we expected to find significant differences in the metabolomic profiles at this age, identifying critical metabolites participating in the increased lifespan. Instead, the differences were more robust at 30 days, suggesting that some anti-aging signals could occur at this age before differential survival is evident.

For this reason, we focused our efforts on identifying significant metabolomic differences at the 30 days stage, and then tracked the dynamics of these specific metabolites with age. To achieve this, we built heat maps that showed the 20 metabolites that had a more significant contribution to the separation of the genotypes in each anatomical section (Figure 4).

A wide range of metabolites changed between wildtype and KD’s, including carbohydrates, amino acids, lipids, organic acids and nucleotides; many of which had already been linked to age-related changes. Although the clustering by genotype (top tree branching) is not perfect, in general, changes in *Marf^KD^* and *Opa1^KD^* compared with the control were similar. We did not observe this trend in *Opa1^KD^* abdomen, where no clear separation from the control flies was found. Interestingly, the trend clustering (left tree-branching) showed clearly that the same metabolites tended to cluster together in most trees. In support of this, correlation analysis (Appendix A) revealed that several metabolic changes seemed to be always positively correlated. Three of those clusters contained for example arginine, betaine, carnitine, and sugars; fumarate, formate and nucleotides; branched-chain amino acids leucine, isoleucine and valine.

It is remarkable that alterations in metabolites related to carbohydrate metabolism were common in all the genotypes and body regions. In general, we observed an upward trend in carbohydrates, such as glucose and glycogen within the *Marf^KD^* and *Opa1^KD^* genotypes compared with the control flies in both head and thorax tissues. These differences were more pronounced in the head, suggesting a high non-autonomous effect caused by *Marf* and *Opa1* downregulation in the muscle tissue (see below). Trehalose was the only carbohydrate whose levels were reduced instead of increased in the KD flies. Although glucose and trehalose are circulating sugar metabolites, their levels can be inversely correlated as they are controlled by different *Drosophila* insulin-like peptides (Dilps) [23,24]. Betaine and carnitine, both related to lipid metabolism, were also generally upgraded in the KD samples.

While carbohydrate changes were consistent, other compounds related to energy metabolism had regional differences. Thorax samples of the KD genotypes had increased levels of pyruvate and decreased levels of succinate, related to glycolysis and tricarboxylic acid cycle, respectively. Therefore, in these genotypes muscle seemed to be more glycolytic and less oxidative. The trend for succinate was the opposite in heads and *Marf^KD^* abdomens.

As for aminoacids, we had already observed that there was a trend towards lower abundance with age in branched-chained and aromatic aminoacids (Figure 2 and Appendix A). The heatmap reveals that in younger flies, levels were already lower in RNAi flies than in the control. The dynamics were similar for histidine and aspartate, especially in the head and thorax. A particularly interesting aminoacid is β-alanine, this aminoacid is most abundant in muscle and related with muscular preservation [21]. Its greater presence in RNAi flies could be related to their improved viability and performance in the negative geotaxis assay, we did not observe histological differences in muscle integrity though (Figure 1D). *N*-acetylaspartate was also consistently decreased in all the samples of the KD genotypes. This metabolite is synthesized in the mitochondria of the neurons but can be also released into the system. In humans, it is considered as a biomarker for neuronal health, but its biological significance is still unclear [25].

### 2.5. The Metabolomic Changes Related to Age Are Anticipated in the Knock-Down Genotypes

The previous analyses showed that particular classes of metabolites had significant changes with age, but also between the two RNAi genotypes and the control in young flies. We chose those metabolites that contributed most to these differences to study how their dynamics by age and anatomical region are modified in the knock-down genotypes (Figure 5).

Regarding the aminoacids, there was a general trend towards lower abundance with age. This was true for histidine, the aromatic aminoacid tyrosine, the branched-chain aminoacids leucine and valine, aspartate and *N*-acetyl-aspartate. Interestingly, what we observed in the *Marf^KD^* and *Opa1^KD^* flies was not a preservation of the levels in the older flies, but an anticipation of the change in the younger flies. The differences were not always statistically significant in all comparisons, but the trend was common to all the comparisons. An exception to this trend was β-alanine, which was much more abundant in thorax, and its high levels were better preserved in the older knockout flies. As we have mentioned, this metabolite is linked to muscle health, which indicates that this tissue is protected from aging and validates our assumption that the thorax extracts are mostly representative of autonomous interference in the muscle.

The magnitude of the changes of carbohydrates was much more pronounced than that of other kinds of metabolites, which indicates that an alteration of carbohydrate metabolism might be directly related to the increased lifespan and preserved locomotor capacities of the *Marf^KD^* and *Opa1^KD^* flies. Glucose and glycogen tended to increase with age, and this increase was also anticipated in RNAi genotypes. In contrast, trehalose tended to decrease in control flies, while RNAi flies showed lower levels at an early age. In insects, trehalose, not glucose, is the main circulating sugar, so low trehalose would be indicative of a caloric restriction with age, which again is anticipated in young knockout flies. Interestingly, two of the metabolites with most significant increases with age and in the knock-down genotypes, betaine and carnitine, are also important in energetic metabolism: carnitine is a transporter of fatty acids and helps to mobilize them, in humans and animals, high levels of betaine are also associated with fat catabolism [26,27].

Energetic nucleotides (ATP and ADP) also tended to decrease with age, and this decrease was anticipated in younger knockout flies, especially in thorax and abdomen. Looking at the metabolic changes in detail, it is noteworthy that many changes between wildtype and KD genotypes had a similar trend than age-related changes. This was true both for those metabolites that reduced their levels (aminoacids, trehalose, energetic nucleotides) and for those that increased them (trehalose, betaine, carnitine). So, what we observed was that down-regulation of *Marf* and *Opa1* partially anticipated or mimicked the changes that occurred with aging. This dynamic explains why the differences between the control and RNAi genotypes were more significant at 30 days than at 65 days.

### 2.6. Interference with Marf and Opa1 in the Muscle Induces a Non-Autonomous Down-Regulation of the Insulin Pathway

The interference with these two mitochondrial fusion-promoting genes in the muscle induced an extension of the lifespan and a reshaping of the metabolome, especially the carbohydrate profile. It has been already described that those physiological alterations of the muscle tissue can extend lifespan through a non-autonomous effect involving down-regulation of the *Drosophila* insulin pathway [11,12]. In fact, mammalian *Mfn2* has already been linked to insulin signaling and carbohydrate metabolism [28]. The insulin signaling pathway is conserved in *Drosophila* and other invertebrates. The ligand of the insulin receptor, instead of insulin and IGF, are the *Drosophila* insulin-related peptides (Dilps) secreted by certain neurons in the central nervous system and other tissues involved in metabolic control such as the fat body [10]. The genes coding for the insulin receptor and the translation initiation factor *4EBP* are direct transcriptional targets of FOXO, which is down-regulated through Akt upon insulin pathway activation. Therefore, these are used as reporters of the down-regulation of the insulin pathway [29].

We determined transcript levels of *InR* and *4EBP* in extracts of head, thorax, and abdomen of 30-day old flies as an indicator of the level of activation of the insulin pathway in *Marf^KD^* and *Opa1^KD^*, compared to the control flies (Figure 5F). Surprisingly, the different anatomical samples had different levels of activation. While the insulin pathway seemed to be more activated in the thorax, it was down-regulated in head and abdomen extracts. Translating these results into body region-specific responses, blocking mitochondrial fusion in the muscle would produce an autonomous activation of the insulin pathway and a non-autonomous down-regulation of this pathway in the rest of the organism. This down-regulation could explain both the metabolic changes and the lifespan extension.

### 2.7. Lipid Changes in Head Extracts

Changes in carbohydrate metabolism are normally associated to alterations in lipids, since they are alternative sources of cellular energy. Head extracts had the highest complexity in lipid composition, contributing to their separation from the other anatomical regions (Figure 1E,F). To obtain more details about non-autonomous changes related to lipids, an LC-MS-based lipidomic study was performed by analyzing the organic phase of the head extracts. This technique allows for a better resolution of the complex lipid species.

First, we identified which lipids had a significant change between young and old heads within each genotype. Those metabolites belonging to the same species and showing the same tendency were grouped (sum). We observed lipid changes in the three genotypes (Figure 6A–C). Among the lipids with significant differences, the general trend was a decrease in abundance with age, regarding both the number of species and the magnitude. Additionally, there were more changes in both knock-down genotypes compared to the control.

The lipids that tended to decrease with age included most mono- and poly-unsaturated lipids. Only unsaturated Sph(18:1) increased with age in control but not in *Marf^KD^* and *Opa1^KD^*, while unsaturated lysoPE(18:3) increased only in *Marf^KD^*. The lipids that increased with age included several saturated fatty acids (FA) and in the case of *Marf^KD^* also saturated fatty acid esters of hydroxy fatty acids (FAHFA). Interestingly, several saturated mono- or triglycerides (MG(16:0), MG(17:0) and MG(22:0) in *Marf^KD^*; TG(46:0), TG(47:0), MG(22:0) and MG(18:0) in *Opa1^KD^*) were increased with age in the modified genotypes, but not in control samples.

Next, we compared both of the knock-down genotypes to the control at 30 and 65 days (Figure 6D–G). In general, the trend was similar to that observed in the previous NMR analysis, the control had more robust differences to *Marf^KD^* than to *Opa1^KD^*. In both cases, the number of lipid species that were significantly different was higher in younger flies (Figure 6D,F) and the difference became less prominent with age (Figure 6E,G). Interestingly, in both KD genotypes (*Marf^KD^* than to *Opa1^KD^*), an increase in several saturated DG and TG was detected at both ages. Furthermore, higher levels of two unsaturated phospholipids (PG39:1 and PG41:1) could be observed only in *Opa1^KD^.*

## 3. Discussion

This work shows that reduced mitochondrial fusion in the muscle has beneficial anti-aging effects evident in an extend lifespan and improved locomotor capacities. Additionally, we demonstrate, for the first time, a link of muscle mitochondrial fusion perturbation with the alteration of the systemic metabolome reorganization that takes place in the aging process. Finally, we observed a transcriptional response of insulin pathway marker genes that would favor carbohydrates synthesis and lifespan extension. Previous metabolomic studies have found a correlation between metabolomic changes during aging. For example, dietary restriction extended lifespan by altering specific metabolites [30,31]. Our work also builds upon previous observations that local, muscle-specific, alterations can affect longevity [11,32] and confirms some of their observations. In addition, our study adds an extra layer of novelty by taking into account the non-autonomous metabolic effects caused by local alterations only in the muscle.

Principal component analysis revealed that there are specific metabolic profiles related to age (30 vs. 65 days of age). Regarding the anatomical regions, we used them to estimate autonomous (thorax) versus non-autonomous (head, abdomen) metabolomic changes. We were aware that there were factors that would introduce noise in this estimation. First, the anatomical regions are not histologically pure. The head can contain tissues other than nervous, although this is predominant; the thorax certainly contains some central nervous system (the thoracic ganglion) and digestive tissue, and the abdomen is histologically complex. Second, some metabolites can freely diffuse in the hemolymph regardless of their origin. Despite these caveats, the three regions have distinctive metabolic signatures consistent with our assumptions: higher diversity of lipids in the head and predominant presence of β-alanine in the thorax. 

Surprisingly, the knock-down and the control genotype’s differences were more evident in the younger flies in all three anatomical regions. There was high variability in all genotypes at 65 days, making it more challenging to separate them. One factor that could contribute to blur the differences at 65 days of age would be a more complex metabolome. In fact, it has already been determined that metabolite diversity increases with age [30]. For most metabolites, basal levels in younger knock-down flies tended to resemble the state of older flies. These results suggest that down-regulation of both genes produces metabolic changes that are protective towards aging. Therefore, the anti-aging effect of mild mitochondrial disturbance in the muscle could be determinant in the first 30 days of *Drosophila* life. In agreement with this, it has already been described that mitochondrial fission induction in a narrow window in midlife (30–37 days) extends *Drosophila* lifespan and improves climbing capacity, among other benefits [13]. This induction was performed ubiquitously; in contrast, we performed mitochondrial dynamics perturbation only in muscle.

The relationship of aminoacids with aging and life extension has been extensively explored. Dietary restriction-mediated lifespan extension in *Drosophila* is largely due to the low abundance of aminoacids, including essential aminoacids, with a particular requirement of methionine [33] but with additional involvement of other essential and non-essential aminoacids including BCAAs, lysine, threonine and histidine [34,35]. A decrease in aspartate and glutamate has also been related to and extended life expectancy obtained by heat treatment [36]. Consistent with this, we have observed a reduction in several aminoacids already in young flies, which would be protective against aging.

The most remarkable changes were in metabolites involved in energy and carbohydrate metabolism. Trehalose was reduced in the head, thorax and abdomen, while glucose, myo-inositol, and glycogen presented significantly higher levels than control flies in all regions. In *Drosophila*, the main circulating carbohydrate is the disaccharide trehalose, while glucose concentration in the hemolymph is much lower and glycogen storage occurs predominantly in the muscle [37]. Trehalose is much more abundant than glucose in the hemolymph and is the primary fuel for high demand energy tissues such as indirect flight muscles and neurons. Moreover, trehalose and glucose levels are controlled independently: glucose levels respond to food composition, while trehalose levels are kept within a narrow concentration range [38]. Therefore, the reduced trehalose levels in all tissues and ages in the knock-down flies would result in a fuel availability reduction, mimicking a caloric restriction effect. 

The lower levels of aminoacids and trehalose would be akin to a systemic dietary and caloric restriction. The main candidate to mediate this metabolic state is the insulin signaling pathway. Our quantitative PCR analyses of the insulin pathway activation markers *InR* and *4EBP* revealed that down-regulation of *Marf* and *Opa1* in the muscle would result in an autonomous activation of the insulin pathway, and in a non-autonomous down-regulation of the same pathway in the rest of the organism. This non-autonomous effect is more evident in the head.

In contrast with the situation we describe here, Demontis and Perrimon reported that increasing FOXO and *4EBP* activity in muscle promotes proteostasis and triggers a systemic insulin down-regulation that increases lifespan [12]. Therefore, from opposite situations regarding the levels of FOXO and *4EBP* activity in the muscle, we reach a similar response. At this point, and in light of the available evidence, it is not possible to find an explanation to this apparent contradiction. It is possible that the cause is that the genetic interventions in both works are different; in our case, we did not alter FOXO or *4EBP* directly, we interfered with mitochondrial fusion, which may have other effects in addition to those in these two proteins. An alternative explanation would involve the role of hormesis, a response to mild and non-specific stress, which could elicit a protective response in the whole organism and an extended lifespan [39].

The insulin pathway has an effect on the balance between glycolysis and the tricarboxylic acid cycle. In agreement with this, we have detected changes in the thorax that are consistent with a down-regulation of the TCA and an increased glycolysis: decreased levels of succinate and increased levels of pyruvate. In the rest of the organism, the down-regulation of the insulin pathway would have the opposite effect, a reduction in the mitochondrial function that has been shown to be closely related with aging [40]. Succinate, which is increased in the head extracts, is one the key metabolites in mitochondrial activity and in aging cells there is a decline in transcript expression of succinate dehydrogenase [41]. It has been further hypothesized that succinate has the potential to restore the loss in functions associated with cellular senescence and systemic aging [42]. Thus, the increased levels of succinate in *Opa1* and *Marf* genotypes in those regions where this metabolite is present in a higher abundance could be related with increased life expectancy.

Levels of carnitine and betaine tended to be also elevated in the knock-downs. In patients and animal models, betaine deficiency is associated with metabolic disorders and diabetes [26]. The role of carnitine is the transport of fatty acids through the mitochondrial membrane and the buffering of Coenzyme A concentration in the mitochondrial matrix, and therefore it is involved in metabolic control [27]. Carnitine could mobilize the lipids for their use as fuel by the mitochondria, thus preventing their accumulation, which can be detrimental for insulin signaling. Therefore, high carnitine levels are linked to better glucose utilization and the accumulation of glycogen. In the case of *Drosophila*, the circulating sugar is trehalose rather than glucose.

Regarding the lipids, NMR revealed an age-related increase in triacylglicerols and phospholipids and decrease in MUFA/PUFA, but the most evident changes were in head extracts: increased esterified lipids and a marked reduction in unsaturated lipids. The MS analysis of head extracts confirmed that most mono- and poly-unsaturated lipids decrease with age, while several saturated fatty acids have the opposite trend. Interestingly, particular classes have a genotype-specific behavior such as increased unsaturated Sph (18:1) in control flies, increased unsaturated lysoPE(18:3) and saturated fatty acid esters of hydroxyl fatty acids (FAHFA) in *Marf^KD^*, which suggest differences in age-related lipidic metabolism in the three genotypes Also saturated mono- or triglycerides increase with age only in the knock-down genotypes.

When we compared each of the knock-down genotypes to the control, we found the features were similar to our previous NMR analyses: the number of lipid species with significant changes was much higher in younger flies, and that *Marf^KD^* had more robust differences than *Opa1^KD^*. Compared to the control, *Marf^KD^* and *Opa1^KD^* had increased levels of saturated di- and triglycerides. These kinds of neutral lipids are the main constituent of lipid droplets that are cellular lipid storage organelles [43]. Thus, it seems that the knock-down genotypes have higher levels of lipid storage in head that increases with age. Already previous studies have detected an increased lipid accumulation as a result of a genetic intervention that results in a lifespan extension [44], or as a result of a downregulation of insulin signaling [45].

There are several lines of evidence for a relationship of mitochondrial function and mitochondrial control of metabolism and aging in *Drosophila* and mammals [46,47,48,49,50,51]. All these works revealed that ubiquitous manipulation of mitochondrial physiology presents significant advantages for longevity. In turn, we obtained lifespan benefits modulating mitochondrial dynamics only in muscle, and it is in line with the proposed idea that muscle is a crucial organ participating in systemic aging [32]. Interestingly, humans’ mortality rate and age-related diseases are closely related to muscle fitness [52,53]. Furthermore, our results can be relevant for other diseases as the understanding of *Marf* and *Opa1* functions could reveal new pathways for Charcot-Marie-Tooth disease and optic atrophy, caused by *MFN2* and *OPA1* mutations, respectively [54,55].

## 4. Materials and Methods

### 4.1. Drosophila Stocks and Culture

We obtained the following *Drosophila* stocks from the Bloomington *Drosophila* Stock Center: The *UAS-RNAi* lines were B#31157 (*Marf*) and B#32358 (*Opa1*), both from the TRIP collection, *Mhc-Gal4* (B#38464) and *UAS-Dcr-2* (B#24650). The wildtype strain was *Oregon-R*. For the RNAi experiments we synthesized a stock with *Mhc-Gal4* on chromosome II and *UAS-Dcr-2* on chromosome III. For the control flies, this stock was crossed to *Oregon-R*, for the experimental flies it was crossed to each *UAS-RNAi* stock. The flies were kept in standard corn meal medium, and all the experiments were performed at 25 °C.

### 4.2. Lifespan Assay

A total of 15–20 newly eclosed flies from each genotype were placed in a vial and kept in a vial at 25 °C. Every 2–3 days the number of deaths was recorded, and the survivors were transferred to a fresh vial. The results are the average of five replicates per genotype. The statistical analyses of lifespan were calculated by Log-rank (Mantel-Cox) tests in GraphPad Prism 8.

### 4.3. Negative Geotaxis Assay

The flies were anesthetized under CO_2_, and 15 flies of each genotype were put in an empty vial, with a mark at 90 mm from the bottom, and were allowed to recover for 30 min. The flies were knocked down to the bottom of the vial with a vigorous tap and recorded on video. The videos were analyzed to determine the number of flies that had crossed the mark after 10 s. The assays were performed at 10, 30 and 60 days, with three replicates per genotype.

### 4.4. Histology

For microscopy, six thoraces of each genotype were dissected and fixed with 2% paraformaldehyde, 2% glutaraldehyde; post-fixed in OsO_4_ for two hours and dehydrated in an ethanol series. Finally, the thoraces were embedded in epoxi resin (Durcupan, Sigma-Aldrich, Saint Louis MO, USA). For bright field microscopy, semi-thin 1.5 µm sections were stained with toluidine blue and images were examined with a Leica DM6000 microscope (Leica, Wetzlar, Germany). For transmission electron microscopy, 80 nm thick sections were stained with uranyl acetate and examined with a FEI Tecnai Spirit G2 microscope (TEI, Hillsboro, OR, USA).

### 4.5. Quantitative PCR

RNA was extracted from the three anatomical regions dissected from 30 flies following the Trizol method. Reverse-transcription was made with qScript cDNA supermix (Quanta Biosciences). PCR primers were obtained using the Universal Probe Library Assay Design Tool (Roche, Basel, Switzerland): *Opa1* forward, 5′-CATGGCACACTACTTTTCCTGA-3′, and reverse, 5′-TGCTACTAGCCGAGGAGCTAAT-3′, *Marf* forward 5′-CGCCAGTTGTTTGATGTTCA-3′, and reverse, 5′-ATTGGGCACACCACGAAT-3′, *4ebp* forward 5′-CCAGATGCCCGAGGTGTA-3′, and reverse, 5′-AGCCCGCTCGTAGATAAGTTT-3′, *InR* forward 5′-CCGGGATTACTGTACTGAACCT-3′, and reverse, 5′-CGCCTGCTAAAGGATCTGA-3′. Expression levels were normalized by the expression of *Rp49* with the following primer sequences: *Rp49* forward, 5′- CGTTTACTGCGGCGAGAT-3′, and reverse, 5′-GCGCTCGACAATCTCCTT-3′. Quantitative real-time PCR was performed with SYBR Green from FastStart Essential DNA Green Master (Roche). Ct values were obtained using a real-time quantitative PCR system LightCycler 480 (Roche). Each sample was analysed in triplicate, and the expression was calculated according to the 2^−^^ΔΔ^^Ct^ method. Agarose gel electrophoresis was also performed with all the PCR products to check that the amplicon size was correct and to rule out non-specific amplification.

### 4.6. Metabolite Extraction for Nuclear Magnetic Resonance

Metabolite extraction was performed with the chloroform-methanol-water extraction procedure, optimized previously for *Drosophila melanogaster* [56]. First, 20 flies from each genotype (Control, *Marf^KD^* and *Opa1^KD^*) and age (30 and 65 days) were dissected into three body regions (head, thorax and abdomen). Each part was placed in a plastic tube on ice and homogenized in a solvent made with 240 μL methanol and 120 μL chloroform. After homogenization, 120 μL of chloroform and 120 μL of water were added to the samples, and vortexed again. Then, samples were kept 15 min on ice and centrifuged at 4 °C for 15 min at 10,000× *g*. At this point, two phases were obtained: an aqueous phase at the top (polar metabolites) and a chloroform phase at the bottom (lipophilic metabolites). The two phases were separated in different tubes. Then the aqueous phase was lyophilized while the chloroform of the organic phase was evaporated under vacuum. Samples were stored at −80 °C. Each experiment was repeated six times.

### 4.7. Nuclear Magnetic Resonance Sample Preparation and Spectroscopy

Aqueous extracts were resuspended in 550 μL of NMR buffer (100 mM phosphate, in D_2_O, pH 7.4, 0.1 mM 3-(trimethylsilyl)-2,2′,3,3′-tetradeuteropropionic acid (TSP)). Samples were then transferred in a 5 mm NMR tube. Organic extracts were resuspended in 600 μL deuterated chloroform with 0.0027% *v/v* Tetramethylsilane (TMS) and transferred to an NMR tube. Samples were stored at 4 °C and measured the same day.

NMR spectra of the extracts were recorded at 27 °C on a Bruker AVII-600 spectrometer using a 5 mm TCI cryoprobe. One-dimensional ^1^H-NMR noesy spectra were acquired with 512 free induction decays (FIDs) for the aqueous extracts and 256 FIDs for chloroform extracts. In total, 64k data points were digitalized over a spectral width of 30 ppm for extract spectra. A 3 s relaxation delay was included between FIDs and water presaturation was applied for aqueous samples. The FID values were multiplied by an exponential function with a 0.5 Hz line broadening factor for extracts and 1 Hz for intact cells. Total Correlation Spectroscopy (TOCSY) and multiplicity Heteronuclear Single Quantum Correlation (HSQC) were performed for representative samples of the aqueous phase and the organic phase of each condition for signal assignment. For each of these experiments, 256–512 t1 increments were used and 32–96 transients were collected. The relaxation delays were set to 1.5 s and the experiments were acquired in the phase-sensitive mode. For extracts, TOCSY spectra were recorded using a standard MLEV-17 pulse sequence with mixing times (spin-lock) of 65 ms. For the extracts, the TOCSY spectra were recorded using a standard MLEV-17 pulse sequence with mixing times (spin-lock) of 65 µs.

### 4.8. NMR Data Analysis

Metabolite identification and assignment was performed with the help of previous data [56], with information from 2D NMR experiments and the databases Amix, HMBD [57] and Biological Magnetic Resonance Data Bank. For metabolite quantification, spectra were automatically integrated at selected regions with MestreNova with GSD deconvolution. For a better comparison between extract samples, integration values were normalized to total intensity to minimize the variability generated by the extraction procedure.

Multivariate data analysis was performed with SIMCAP 14.1 (Umetrics, Umeå, Sweden) with the normalized integral values of polar and nonpolar metabolites. Integral tables were univariate scaled (each value is divided by the standard deviation of each variable) for an easier interpretation of the data and to take into account small signals. Additionally, mean centering was applied to improve the interpretability of the model. Principal components analysis (PCA) was used as an unsupervised model. PCA score plots show clustering trends between samples and if there were any strong outliers. Loading plots were used to relate these trends with metabolite concentrations. For discriminant model building, OPLS-DA (orthogonal partial least square discriminant analysis) was performed. To assess the quality of each model, the parameters R2 (goodness of fit) and Q2 (goodness of prediction) were evaluated. A Q2 value ≥ 0.5 was considered indicative for a good model. OPLS-DA models were validated by permutation (*n* = 20) and ANalysis Of VAriance testing of Cross-Validated predictive residuals (CV-ANOVA).

Heat maps were performed with Metaboanalyst [58] using ANOVA, a Ward Clustering Algorithm and a Euclidean Distance Measure.

### 4.9. Sample Preparation for Mass Spectrometry

The procedure to obtain the organic head extracts was the same as in as in the NMR analysis, in this case each experiment was repeated three times. Lipids were extracted by adding 150 µL of cold isopropanol to each dried sample, vortexed, and kept at −20 °C for 30 min for protein precipitation. After centrifugation at 13,000 *g* (10 min, 4 °C), 90 µL of the supernatant were transferred to a 96-well plate and 10 µL of an internal standard mix solution (IS, 20 µM), containing lipids from different classes, were added to each sample. A quality control (QC) sample was prepared by mixing 10 µL of each sample. A blank was prepared to identify potential artefacts. Finally, samples, were randomly injected in the chromatographic system in order to avoid intra-batch variability, as well as to enhance quality and reproducibility. One QC Sample was injected every 5 samples, and the blank was injected at the end of the analytical sequence. Stability and analytical drift were investigated through IS intensities. 

### 4.10. UPLC-QToF Analysis

Samples were analysed on an Ultra-Performance Liquid Chromatography (UPLC) system coupled to an iFunnel quadrupole time of flight (QTOF) Agilent 6550 spectrometer (Agilent Technologies, Santa Clara, CA, USA). The chromatographic separation of lipids was carried out using an Acquity UPLC C18 CSH chromatographic column (100 × 2.1 mm, 1.8 µm) (Waters, Wexford, MA, USA). The UPLC-MS method employed was previously described by Alcoriza et al. (2019). Briefly, for ESI(+) mode, the mobile phases consisted of (A) 10 mM ammonium formate in 60:40 (*v/v*) acetonitrile:water and (B) 10 mM ammonium formate in 90:10 (*v/v*) isopropanol:acetonitrile and a flow rate of 0.4 mL/min; for ESI(−) mode, ammonium acetetate was used as modifier, and the flow rate employed was 0.6 mL/min. Autosampler and column temperatures were set to 4 °C and 65 °C, respectively and the injection volume was 5 µL.

Samples and QC were acquired using Full scan MS data from 50 to 1700 *m/z* with a scan frequency of 6 Hz. QC were also acquired using dependent data acquisition (DDA), by Auto MS/MS mode, and data independent acquisition (DIA), by using the all-ion fragmentation mode, both using 0 and 40V as collision energies.

### 4.11. Mass Spectrometry Data Pre-Processing and Analysis

Data processing of samples and QC acquired in Full MS Scan was done by using an in-house R (v.3.6.1) processing script with XCMS and CAMERA packages for peak detection, noise filtering and peak alignment. QC data acquired by DIA mode were processed and annotated by using the LipidMS package [59]. Parameters selected for peak peaking and peak grouping were based on previous instrumental data analysis experience. The resulting data matrix was generated, including annotated molecular features, sample ID (observations) and peak intensities. The dataset was then normalized by LOESS regression (locally estimated scatterplot smoothing) and filtered according to the quality assurance criteria of coefficient of variation < 30% in QC samples and the presence of the variable in 60% of the samples in at least one of the compared groups.

Once the data table was obtained with potential lipid identification, a Volcano Plot was built, and those lipids for which the analytical response changed significantly (FC < 2, *p* value *t*-test < 0.01) in *Marf^KD^* and *Opa1^KD^* compared to the Control at 30 and 65 days were selected. The differences in lipids between 30 and 65 days were also evaluated independently for each group. In order to facilitate the data interpretation, some lipids belonging to the same species were summed as long as they varied the same way.

## Figures and Tables

**Figure 1 ijms-22-12133-f001:**
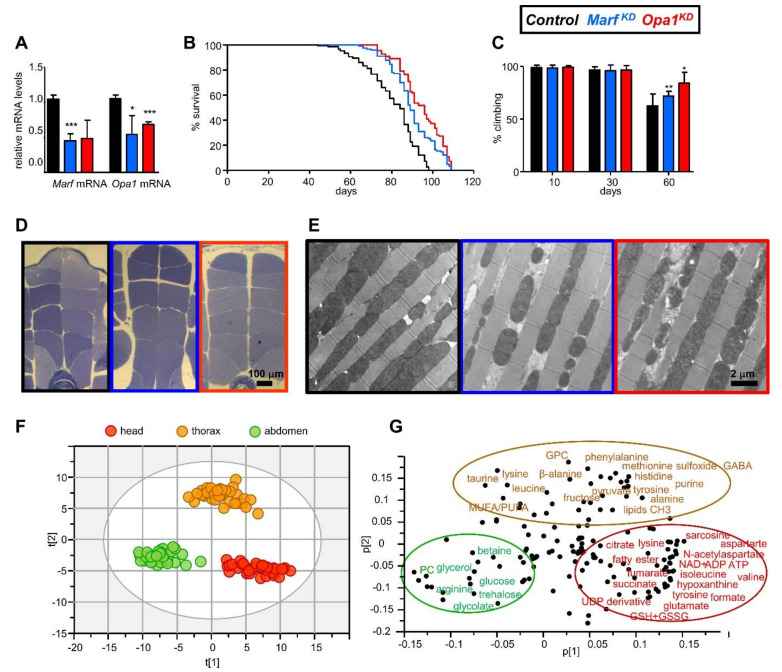
Characterization of the control, *Marf^KD^* and *Opa1^KD^* genotypes; and the metabolomic features of the body regions. (**A**) Transcript levels of *Marf* and *Opa1* in thorax extracts in control *Marf^KD^* and *Opa1^KD^* genotypes estimated by qPCR. (**B**) Survival curves from the three genotypes show statistically significant differences between the control and the RNAi flies (Mantel-Cox test). (**C**) Locomotor capacities studied by negative geotaxis show statistically significant differences at 60 days. (**D**) Thorax semi-thin sections to analyse structural integrity of the indirect flight muscles at 65 days. (**E**) Transmission electron micrographs of longitudinal sections of the indirect flight muscles. (**F**) PCA analysis of the metabolome of the anatomical regions, UV scaling; score plot, 1 + 2 components; R2X (cum) = 0.45; Q2 (cum) = 0.42. (**G**). Loading plot to identify the metabolites that contribute to the separation. GPC: glycerophosphocholine; PC: phosphocholine; MUFA/PUFA: mono-unsaturated/poly-unsaturated fatty acids; GABA: gamma-aminobutyric acid; GSH + GSSG: reduced and oxidized glutathione. In bar diagrams, * *p* < 0.05, ** *p* < 0.01; *** *p* < 0.005.

**Figure 2 ijms-22-12133-f002:**
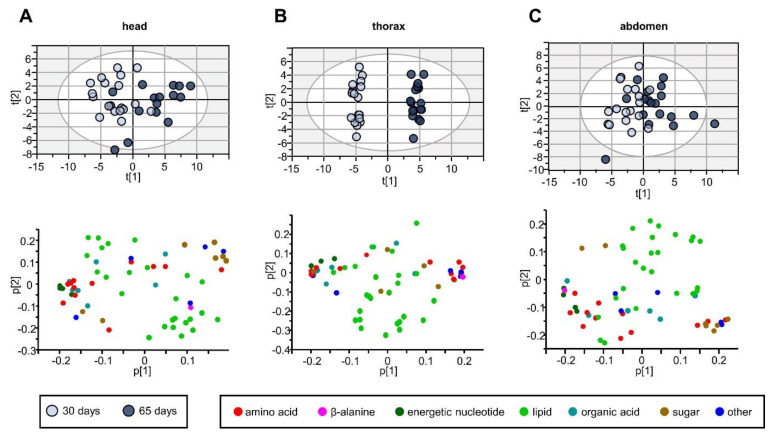
PCA and loading plot of the 30– and 65–day extracts of the three anatomical regions. (**A**) PCA score and plot of head extracts, UV scaling, 6 components, head R2X (cum) = 0.71, Q (cum) = 0.40. (**B**) PCA score and loading plot of thorax extracts, UV scaling, 6 components, head R2X (cum) = 0.72, Q (cum) = 0.47. (**C**) PCA score and loading plot of abdomen, UV scaling, 6 components, head R2X (cum) = 0.72, Q (cum) = 0.47.

**Figure 3 ijms-22-12133-f003:**
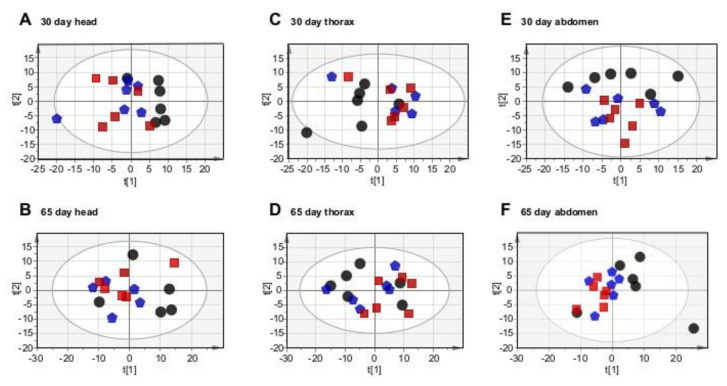
PCA analysis, UV scaling by anatomical region and age. (**A**) PCA score plot of head extracts (30 days), 6 components, head R2X (cum) = 0.67, Q (cum) = 0.25. (**B**) PCA score plot of head extracts (65 days), 6 components, head R2X (cum) = 0.61, Q (cum) = 0.27. (**C**) PCA score plot of thorax extracts (30 days), 3 components, head R2X (cum) = 0.9, Q (cum) = 0.28. (**D**) PCA score plot of thorax extracts (65 days), 4 components, head R2X (cum) = 0.61, Q (cum) = 0.37. (**E**) PCA score plot of abdomen extracts (30 days), 3 components, head R2X (cum) = 0.64, Q(cum) = 0.29. (**F**) PCA score plot of abdomen extracts (65 days), 4 components, head R2X (cum) = 0.65, Q (cum) = 0.20.

**Figure 4 ijms-22-12133-f004:**
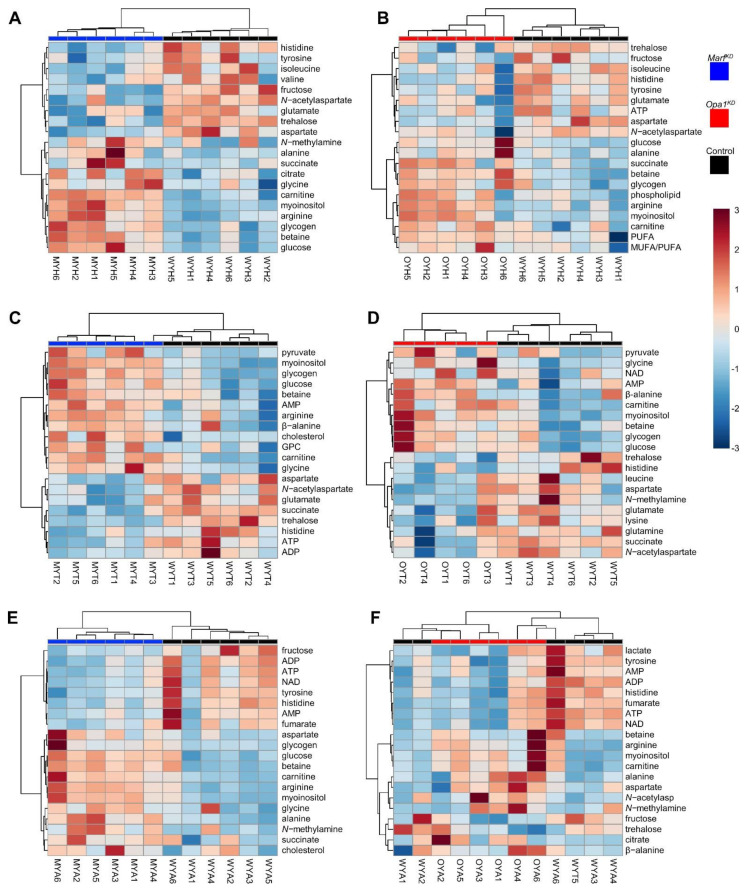
Heatmaps, 20 more relevant features at 30 days, ANOVA *t*-test. (**A**) Heads, control versus *Marf^KD^*. (**B**) Heads, control versus *Opa1^KD^*. (**C**) Thorax, control versus *Marf^KD^*. (**D**) Thorax, control versus *Opa1^KD^*. (**E**) Abdomen, control versus *Marf^KD^*. (**F**) Abdomen, control vs. *Opa1^KD^*. Key to sample codes: C/M/0 means control/*Marf^KD^*/*Opa1^KD^*; Y means young (30-day old); H/T/M means head/thorax/abdomen; The digit is the sample number.

**Figure 5 ijms-22-12133-f005:**
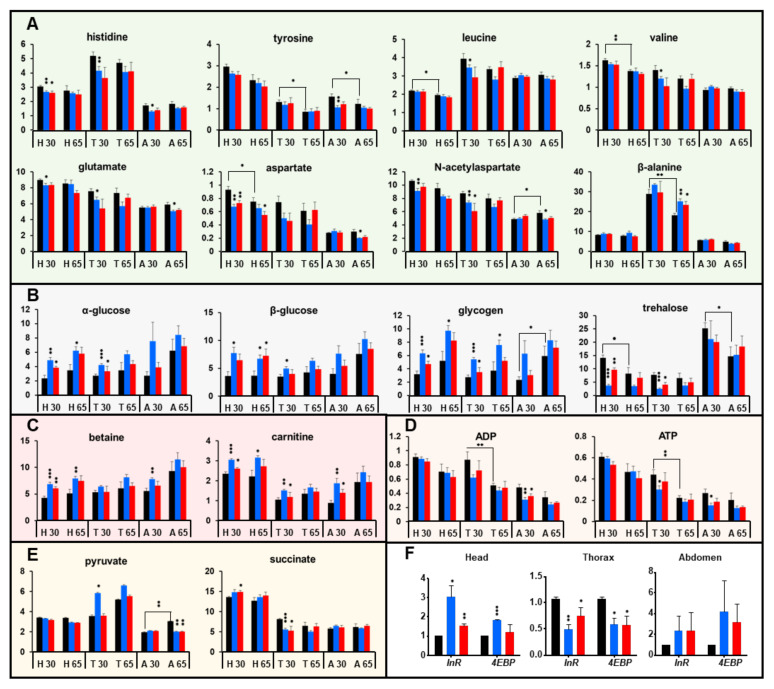
Analysis of metabolites that had significant changes at 30 days by genotype, age and anatomical region. (**A**) Aminoacids. (**B**) Carbohydrates. (**C**) Metabolites related to lipid metabolism. (**D**) Energetic nucleotides. (**E**) Organic acids. (**F**) Relative mRNA levels of the *InR* and *4EBP* genes in head, thorax and abdomen extracts at 30 days. H, T and A refer to the anatomical region; 30 and 65 refer to the age in days. Asterisks over brackets indicate significant differences between young and old control samples; asterisks over colored bars indicate significant differences between KD genotypes and control at that age. In bar diagrams, * *p* < 0.05, ** *p* < 0.01; *** *p* < 0.005.

**Figure 6 ijms-22-12133-f006:**
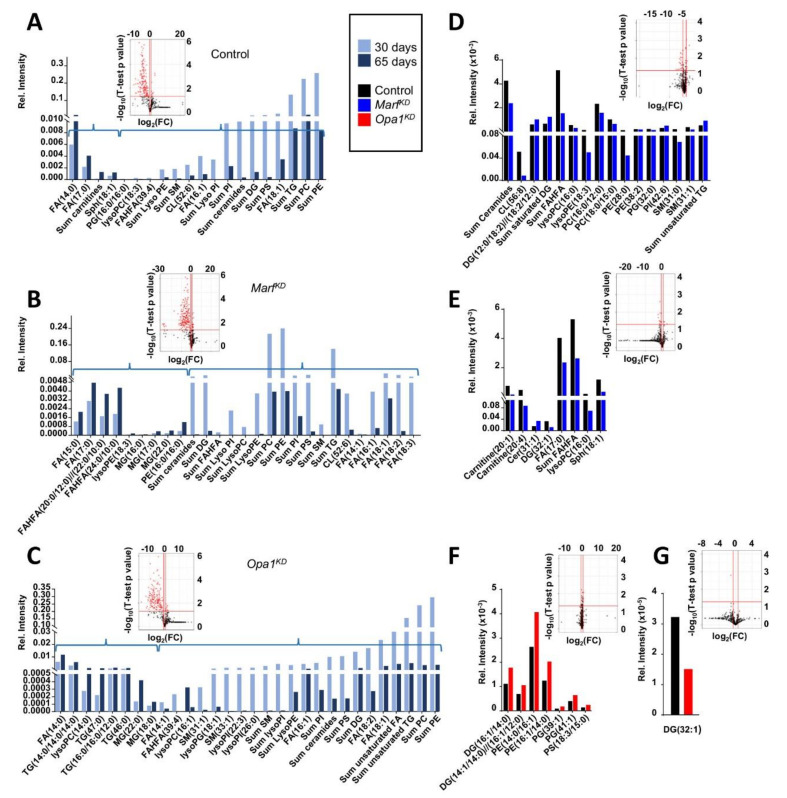
Analysis of head lipids in control, *Marf^KD^* and *Opa1^KD^* flies at 30 and 65 days of age. Volcano plots were built to identify those lipids for which the analytical response changed significantly between the two conditions under comparison (FC < 2, *p* value *t*-test < 0.01). The corresponding bar diagrams compare the abundance of the relevant lipids. In order to facilitate the data interpretation, some lipids belonging to the same species were summed as long as they vary the same way. (**A**) Control 30 vs. 65 days. (**B**) *Marf^KD^* 30 vs. 65 days. (**C**) *Opa1^KD^* 30 vs. 65 days. (**D**) Control vs. *Marf^KD^* at 30 days. (**E**) Control vs. *Marf^KD^* at 65 days. (**F**) Control vs. *Opa1^KD^* at 30 days. (**G**) Control vs. *Opa1^KD^* at 65 days. In **A**–**C**, brackets are used to group the lipids that increase (left) or decrease (right) with age.

## Data Availability

The data that support the findings of this study are openly available in Zenodo at http://doi.org/10.5281/zenodo.5137738 (nuclear magnetic resonance metabolomics) and http://doi.org/10.5281/zenodo.5137758 (mass spectrometry metabolomics).

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
