# Peer review of "Mild Muscle Mitochondrial Fusion Distress Extends *Drosophila* Lifespan through an Early and Systemic Metabolome Reorganization"

_ijms, 2021, doi:10.3390/ijms222212133_

Round 1
Reviewer 1 Report
This is a nice study of the effects of knocking down mitochondrial fusion genes in muscle upon muscle function, survival and metabolic profiles. The manuscript is very well written except for minor grammatical errors that a copy editor should correct, e.g., agreement of noun and verb, using (or not using) “the” in front of a noun. The conclusions regarding lifespan and muscle function are generally sound (see comments below) and the metabolomics, although mostly descriptive, provide detailed comparisons of aging in control and knockdown genotypes for head, thorax and abdomens. The authors should consider the following comments:
1) Knockdown is verified by qPCR, but typically a different second RNAi is utilized to reduce the chance of off-target effects. Have the authors ensured that no such off-target genes exist?
2) For qPCR, did the authors run end-point PCR to show that the expected single band was produced?
3) Lifespans should be indicated as average values and deviations.
4) In Fig 1A, is the Marf mRNA reduction not significant in the Opa1KD? In this regard, please note the meaning of *, **, *** for the readers
5) Low resolution analysis of muscle structure by light microscopy will miss more subtle changes in mitochondrial and myofibrillar architecture. This should be noted.
6) Muscle tissue is found in the head and abdomen, so conclusions of non-autonomous regulation need to be tempered by making this comment in the text.
7) The legend for Figure 4 should explain the meaning of the X axis names: WYA1, OYA6, etc.
8) In the present manuscript, the authors observe down regulation of 4EBP in muscle which would reduce FOXO levels. How does this yield increased lifespan, given the Demontis and Perrimon conclusions (Cell, 2010), which can be summarized in the following quotation: “FOXO/4E-BP activity regulates muscle proteostasis at least in part via the autophagy/lysosome pathway of protein degradation, preserves muscle function, and extends life span. In addition, FOXO/4E-BP signaling in muscles decreases feeding behavior that, similar to fasting, results in reduced insulin release from producing cells. This in turn promotes FOXO and 4E-BP activity in other tissues, preserving proteostasis organism-wide and mitigating systemic aging.”
9) Figure S4 lacks letter designations for each panel (A, B, C, etc.), although they are present in the legend.
Author Response
We thank both reviewers for their time to revise our manuscript and for highlighting the relevance of our work for a broad audience. We also appreciate their comments, which we have used to improve our work. We are providing a point-by-point response to reviewers comments.
Reviewer 2
My single minor concern is that the mild muscle mitochondrial fusion induced by either loss of Marf1 or Opa1 has been inferred by the authors from indirect, although related, evidence. As far as I can tell, Mhc-Gal4>UAS Marf RNAi and Mhc-Gal4>UAS Opa1 RNAi expressing flies have never been characterized before for mitochondrial fragmentation in adult muscles. The authors should therefore provide more direct observations (cytological analysis? TEM?) to show that mitochondria are indeed fragmented in these RNA interfered flies. This will make their conclusion even more consistent with a systemic aging in Drosophila influenced by impaired mitochondrial fusions in muscles.
The new Figure 1 addresses the concern of this reviewer, as we explained in our answer to reviewer 1 and in the text, Marf mitochondria have a markedly reduced size, Opa1 mitochondria had a less marked reduction in size but display the expected cristae morphology.

Reviewer 2 Report
In this paper by Tapia-Gonzales et al, the authors addressed a very interesting metabolomics analysis in flies with an impaired mitochondrial function in muscles. To this aim they depleted Marf1 and Opa1 functions specifically in the muscles tand revealed a non-autonomous systemic metabolome reorganization also in distant tissues (such as head and abomen) that affected Drosophila fitness. Based on these results they postulated the existence of a cross-talk between mitochondrial dynamics and inter-organ communication that could be explored to shed more light on mechanisms underlying muscle aging.
This is a well-written manuscript that could be of interest for a broad readership. My single minor concern is that the mild muscle mitochondrial fusion induced by either loss of Marf1 or Opa1 has been inferred by the authors from indirect, although related, evidence. As far as I can tell, Mhc-Gal4>UAS Marf RNAi and Mhc-Gal4>UAS Opa1 RNAi expressing flies have never been characterized before for mitochondrial fragmentation in adult muscles. The authors should therefore provide more direct observations (cytological analysis? TEM?) to show that mitochondria are indeed fragmented in these RNA interfered flies. This will make their conclusion even more consistent with a systemic aging in Drosophila influenced by impaired mitochondrial fusions in muscles.
